# Sustained Human Hair Follicle Growth Ex Vivo in a Glycosaminoglycan Hydrogel Matrix

**DOI:** 10.3390/ijms20071741

**Published:** 2019-04-09

**Authors:** Sandra Fernández-Martos, María Calvo-Sánchez, Karla García-Alonso, Begoña Castro, Bita Hashtroody, Jesús Espada

**Affiliations:** 1Experimental Dermatology and Skin Biology Group, Ramon y Cajal Institute for Health Research (IRYCIS), Ramón y Cajal University Hospital, 28034 Madrid, Spain; sandy.fernandez@hotmail.com (S.F.-M.); calvosanchezmaria@gmail.com (M.C.-S.); 2Instituto de Investigaciones Biosanitarias, Facultad de Ciencias Experimentales, Universidad Francisco de Vitoria, 28223 Pozuelo de Alarcón, Spain; 3Cantabria Labs, 28043 Madrid, Spain; Karla.garcia@cantabrialabs.es (K.G.-A.); bita.Hashtroody@cantabrialabs.es (B.H.); 4Histocell, Bizkaia Technologic Park, 48160 Derio, Bizkaia, Spain; Bcastro@histocell.com; 5Centro Integrativo de Biología y Química Aplicada (CIBQA), Universidad Bernardo O’Higgins, 8370854 Santiago, Chile

**Keywords:** glycosaminoglycans, hyaluronic acid, WNT/β-catenin, BMP/SMAD, TGFβ, stem cells, human hair follicle

## Abstract

Glycosaminoglycans (GAGs) and associated proteoglycans have important functions in homeostatic maintenance and regenerative processes (e.g., wound repair) of the skin. However, little is known about the role of these molecules in the regulation of the hair follicle cycle. Here we report that growing human hair follicles ex vivo in a defined GAG hydrogel mimicking the dermal matrix strongly promotes sustained cell survival and maintenance of a highly proliferative phenotype in the hair bulb and suprabulbar regions. This significant effect is associated with the activation of WNT/β-catenin signaling targets (CCDN1, AXIN2) and with the expression of stem cell markers (CK15, CD34) and growth factors implicated in the telogen/anagen transition (TGFβ2, FGF10). As a whole, these results point to the dermal GAG matrix as an important component in the regulation of the human hair follicle growth cycle, and to GAG-based hydrogels as potentially relevant modulators of this process both in vitro and in vivo.

## 1. Introduction

Glycosaminoglycans (GAGs) are highly hydrophilic long-chain linear polysaccharides made up of a repeating disaccharide unit, consisting of an amino sugar (N-acetylglucosamine or N-acetylgalactosamine) in combination with a uronic sugar (glucuronic acid or iduronic acid) or galactose [1,2,3,4]. GAGs are usually bound covalently to a protein core forming a proteoglycan molecule and are essential components of the extracellular matrix of most mammalian tissues and of the surface of several cell types [1,3,4,5]. There are six major GAG types, namely, dermatan sulphate (DS), chondroitin sulphate (CS), keratan sulphate (KS), heparin (H), heparan sulphate (HS), and hyaluronic acid (HA; the only GAG type that does not contain sulphate and does not bind to a protein core). Besides their roles as scaffold molecules, GAGs and derived proteoglycans are important players in the control of critical physiological parameters (e.g., the organism water balance) and in the regulation of key cellular functions such as proliferation, adhesion, and migration, and whose specific disfunction is associated with different diseases, including cancer [1,2].

All GAGs are present to some degree in the human skin. The syndecan (containing CS/DS and HS) and glypican (containing HS) proteoglycan types are extracellularly linked to the plasma membrane of basal and suprabasal cells in the epidermal layer. HA is the most abundant GAG in the dermal matrix layer, exerting a basic scaffolding function in combination with the collagen/elastin network [2,3,5]. The dermal matrix also contains different extracellular proteoglycans, including versican (containing CS), decorin and biglycan (containing CS/DS), fibromodulin, lumican, and keratocan (containing KS), and perlecan (containing HS) [2,3,5], that complement the GAG setting of the dermis, supposedly endowing this skin layer with unique functional characteristics. In the human hair follicle, cells in the epidermal part and outer and inner root sheath basal layers express syndecan-1, whereas the dermal part, encompassing the dermal papilla and the hair bulb, contain high amounts of HA, HS, CS, DS, KS, and associated proteoglycans, including perlecan, versican, and biglycan [6].

Considering the reported roles of extracellular or membrane associated GAGs in the regulation of key cellular processes, it can be hypothesized that these molecules may have important roles in the skin, and, particularly, in the hair. Important roles for these molecules have been postulated during skin regeneration and wound healing and in the process of skin aging [2,3,5]. However, despite different reports showing a cyclic expression of GAGs and different proteoglycans during the hair follicle growth cycle [3,5,6], little is actually known at a functional level about the physiological implication of GAGs in the regulation of this biological oscillator. In this context, our aim in this work was to investigate potential stimulatory effects of a defined GAG matrix on the hair follicle growth potential using ex vivo cultured human hair follicles as an experimental system. In this biological model, most cell types in the tissue rapidly cease to proliferate and enter into an apoptotic cell death program after serial culturing in basal conditions (e.g., no growth factors added to culture medium), therefore constituting an adequate system to evaluate the functional effects of GAGs on isolated mini-organs.

## 2. Results and Discussion

Here, we used a unique tunable GAG hydrogel mimicking the dermal matrix composition [7,8], hereafter called HC007, composed of non-crosslinked native GAGs comprising hyaluronic acid of high and low molecular weight and sulfated GAGs (CS, DS, KS, and HS). HC007 behaves as a temporary extracellular matrix substitute in damaged tissues by attracting cells and creating an adequate environment where recruited cells find their innate extracellular matrix to survive, proliferate, attach, and elicit their natural bioactivity [7,8]. Human follicular units (FUs) grown ex vivo, typically containing one or two individualized hair follicles in the growing (anagen) phase embedded in a remnant matrix of fatty and dermal tissues (Figure 1A), were used as the experimental system.

In our experience, this tissue remnant favors optimal hair follicle survival up to days 7–9 in basal ex vivo culture conditions. Most hair follicles in basal conditions typically showed a steady decrease of the hair bulb area, defined here as an area encompassing the hair bulb/dermal papilla and the most intensely pigmented suprabulbar region (Figure 1A–C; see Materials and Methods), a follicular area characterized by high cell proliferation and melanogenesis during the anagen phase ([9] and references therein). Consequently, we found low cell proliferation rates after 6–8 days in ex vivo culture (Figure 1D), followed by extensive induction of apoptosis in hair bulb and suprabulbar regions around days 12–16 (Figure 1E). By contrast, growing hair follicles in HC007 1.5% promoted a sustained and statistically significant enlargement of the hair bulb and suprabulbar regions (Figure 1A–C) associated with the maintenance of a strong cell proliferation phenotype in the tissue after 8 days in ex vivo culture (Figure 1D) and with the absence of significant cell death induction by day 16 (Figure 1E) as compared to control samples. After 16 days in culture, the stimulatory effect of HC007 1.5% appeared as a significant increase in the number of cells (Figure 1C). Additionally, we were able to spot a clearly noticeable, although hardly measurable, increase in the length of the hair shaft in a considerable proportion (about 50%) of hair follicles grown in HC007 1.5% (Figure 1A). These results indicate that a defined GAG hydrogel matrix surrounding an explanted human FU can efficiently support sustained cell proliferation and, presumably, hair follicle growth.

We next quantified the expression of different genes involved in the two major signaling pathways regulating the hair follicle growth cycle: Transforming Growth Factor (TGF)β/Bone Morphogenetic Protein (BMP)/Mothers Against Decapentaplegic Homolog (SMAD), implicated in the maintenance of the quiescence state in the hair follicle stem cell niche and the entrance into the resting (telogen) phase, and WNT/β-catenin signaling, which dictates the activation the stem cell niche and the entrance into the growing (anagen) phase [10]. We found that control hair follicles showed high expression levels of the BMP2 and BMP4 effectors and, concomitantly, of the ID1 and ID2 BMP/SMAD gene targets in the skin (Figure 2A). Interestingly, this expression pattern was not affected by HC007 treatments (Figure 2A). By contrast, key WNT/β-catenin gene targets in the skin, cyclin D1 (CCDN1), a general proliferation marker, and AXIN2, implicated in the metabolic stabilization of β-catenin, were strongly induced by HC007 four days after treatments (Figure 2A). The specific induction of CCDN1 was further confirmed by protein immunolocalization in the tissue (Figure 2B).

In addition, we found that the master intracellular inhibitor of WNT/β-catenin signaling, Glycogen Synthase Kinase- 3 beta (GSK3β), showed high expression levels in control hair follicles and was significantly repressed by HC007 treatments (Figure 2A). Low levels of the WNT/β-catenin extracellular antagonist Dickkopf-related protein 1 (DKK1) were found in control hair follicles, in agreement with previous results in mouse adult hair follicles [11], and this expression pattern was not affected by HC007 treatments (Figure 2A). Interestingly, we also found a statistically significant and specific induction of the expression of TGFβ2 and Fibroblast Growth Factor (FGF)10, factors implicated in the telogen-anagen transition [10], after HC007 treatments, whereas no changes were observed in the expression of FGF7 or Vascular Endothelial Growth Factor (VEGF) (Figure 2A). Finally, we analyzed the expression of the stem cell markers cytokeratin 15 (CK15) and CD34, proteins specifically expressed by skin progenitors after activation of the bulge stem cell niche in the hair follicle and associated with multipotency and self-renewal potential [12,13,14,15,16]. We found that both stem cell markers were significantly activated upon ex vivo growing of hair follicles in HC007 1.5% (Figure 2A). We further confirmed by protein immunolocalization a strong increase in the expression level and in the number of CK15 positive cells all along the outer root sheath in HC007-treated hair follicles (Figure 2C), indicating the activation of the bulge stem cell niche.

The results reported here indicate for the first time that a defined GAG matrix can promote and support the sustained growth of human hair follicles ex vivo. We have previously shown that HC007 behaves as an effective temporary extracellular matrix substitute in damaged tissues by attracting cells and creating an adequate environment where recruited cells find their innate extracellular matrix to survive, proliferate, attach, and elicit their natural bioactivity, also promoting the proliferation and continuous expansion of primary skin cells [7,8]. Here, we have further shown that the stimulatory effect of HC007 in the hair follicle occurs in association with an activation of WNT/β-catenin signaling, bypassing the pervading inhibitory action BMP/SMAD signaling. Interestingly, a specific induction of TGFβ2 is also observed after HC007 treatments, in agreement with the reported functional interaction between different GAGs/proteoglycans and member of the TGFβ superfamily [17,18]. Moreover, it has been reported that WNT/β-catenin signaling during the onset of the anagen phase is accompanied by the specific expression of TGFβ2, that regulates proliferation, differentiation, and extracellular matrix production of dermal fibroblasts [19]. The interplay between WNT/β-catenin and TGFβ/BMP/SMAD to regulate stem cell niche function has been also described in different adult tissues, including the hair follicle [20]. It can be hypothesized that the GAG matrix produced by dermal fibroblasts in response to TGFβ/BMP/SMAD signaling is an essential component in the regulation of the hair follicle cycle and, particularly, in the feedback modulation of WNT/β-catenin signaling. We have also found that the signaling cascade potentially triggered/modulated by the GAG matrix results finally in the activation of the hair follicle stem cell niche, showing up as a significant induction of the CK15 and CD34 stem cell marker and a strong increase in the number of CK15 positive cells. As a whole, these results point to dermal GAGs as an essential component in the regulation of the human hair follicle growth cycle, and to the HC007 hydrogel as a potential modulator of this process both in vitro and in vivo, and as an interesting and useful growth media complement to maintain hair follicle function ex vivo for longer periods.

## 3. Materials and Methods

All methods were performed in accordance with all relevant institutional and European Union (EU) experimental and ethical guidelines.

### 3.1. Ex Vivo Culture of Human Hair Follicles and HC007 Treatment

Hair follicular units (FUs) were obtained from human scalp samples taken from the occipital skin of volunteer donors during routine hair transplant procedures. Eligible patients provided written informed consent, and the Ethical Committee of the Ramón y Cajal University Hospital approved this procedure. Selected follicular units (FUs) typically containing one or two hair follicles and surrounding fatty and dermal tissue remnants were dissected by expert trichologists at the Ramón y Cajal Hospital Dermatology Service. Selected FUs encompassed for the most part hair follicles in the growing (anagen) phase using standard morphological criteria [21]. FUs were grown in Williams E medium (Sigma-Aldrich, St. Louis, MO, USA), supplemented with 10× penicillin/streptomycin, 10× amphotericin B, and 2 mM l-glutamine (all from Gibco Life Technologies, Carlsbad, CA, USA), 5 µg/mL insulin, 5 µg/mL transferring, 20 pM T3 hormone, 0.083 µg/mL cholera toxin, and 0.4 µg/mL hydrocortisone at 37 °C in a 5% CO_2_ humidified atmosphere. Adherent tissue remnants in FUs were maintained throughout the ex vivo growing process to improve hair follicle viability in basal conditions. FUs from the same individual and body location were used in each experimental series.

HC007 was used as a sterile solution of non-crosslinked native GAGs comprising hyaluronic acid of high and low molecular weight and sulfated GAGs (chondroitin sulfate, dermatan sulfate, keratan sulfate, heparan sulfate), with a purity ≥95%, dissolved in a concentration of 15 mg/mL in isotonic solution without preservatives. These native GAGs were obtained, through a patented method (WO 2011/120535), from the Wharton’s jelly of the umbilical cord, a specialized extracellular matrix that permits the generation of a family of biomaterials with regenerative properties. For treatment samples, FUs were grown in supplemented Williams E medium containing 1.5% HC007. Working concentration for HC007 was selected based on previous results in primary cell cultures [7,8]. For control samples, FUs were grown in supplemented basal Williams E medium. Growth medium was replenished every two days in all cases.

### 3.2. Quantification of Hair Bulb Area

For the evaluation of significant morphological changes in vivo in whole hair follicles, high resolution images of FUs growing in 24-well plates were acquired at time 0 after tseeding in the presence or absence of HC007 1.5%, and every 24 h onwards for 16 days, depending on experimental settings. Image acquisition was performed using a Nikon Eclipse Ci LED-fluorescence microscope (Tokio, Japan) using a 2× objective and a 0.55× Reduction Lens adapter coupled to a Jenoptik PROGRES GRYPHAX^®^ SUBRA Super HD camera (Jena, Germany) and suited Version 1.1.8.153 image software pack. We defined the hair bulb area as the hair follicle territory encompassing the hair bulb and dermal papilla as well as the most intensively pigmented area of the suprabulbar region, including associated inner and outer root sheets (see Figure 1). This area is characterized by high cell proliferation and melanogenesis during the anagen phase of the hair follicle cycle [9]. Total hair bulb area in high resolution images of whole hair follicles was spotted and quantified using suited free FIJI software packs (https://fiji.sc/). The fold changes with respect to control samples of means +/- SD of hair bulb measurements in *n* ≥ 10 samples for each experimental condition was represented and *t*-test was used for statistical analysis.

### 3.3. Protein Immunolocalization

To determine target protein localization and expression patterns in hair follicles, histological tissues sections were used. At least one FU in each experimental group was fixed in 3.7% aqueous formaldehyde and embedded in paraffin using standard procedures. Typically, 18–22, 8 µm thick, longitudinal tissue sections were obtained for each FU. As a rule, the most 6-4 central sections, encompassing as much as possible the full length of, at least, the suprabulbar fiber and hair bulb regions, were used for comparative analysis. Antigen retrieval was performed using 10 mM citrate buffer in hydrated sections following standard procedures. Primary antibodies, including anti-cytokeratin 15 (CK15, clon EPR16Y, Abcam Cambridge, UK), and anti-KI67 (clon SP6, Abcam), and anti-cyclin D1 (CCDN1, clon EPR2241, Abcam) were incubated overnight at 4 °C in a wet chamber, extensively washed with PBS, incubated for 1 h at room temperature with appropriate secondary fluorescence- or HRP-labelled antibodies, and mounted in DAPI (100 ng/mL)-containing Vectashield. Confocal images were obtained using a Leica TCS SP5 AOBS spectral confocal microscope (Wetzlar, Germany) and processed using the FIJI software. Bright field images were obtained using a Nikon Eclipse Ci LED-fluorescence microscope (Tokio, Japan) coupled to a Jenoptik PROGRES GRYPHAX^®^ SUBRA Super HD camera.

### 3.4. RNA Extraction and Gene Expression Analysis

Total RNA of at least four FUs in each experimental group was extracted four days after treatments, using an RNeasy micro kit (Qiagen, Venlo, Netherlands). RNA was normalized with respect to the number of FUs in each experimental condition, and then was converted into cDNA using a FastGene Scriptase II cDNA Kit (NIPPON Genetics, Tokyo, Japan). qRT-PCR data was analyzed using a comparative CT method, using 18S ribosomal RNA expression as an internal control. Gene expression fold changes were represented as the ratio between means of 2^−ΔCt^ values of HC007 FU and control FU mean values. Primer sequences are available upon request.

## Figures and Tables

**Figure 1 ijms-20-01741-f001:**
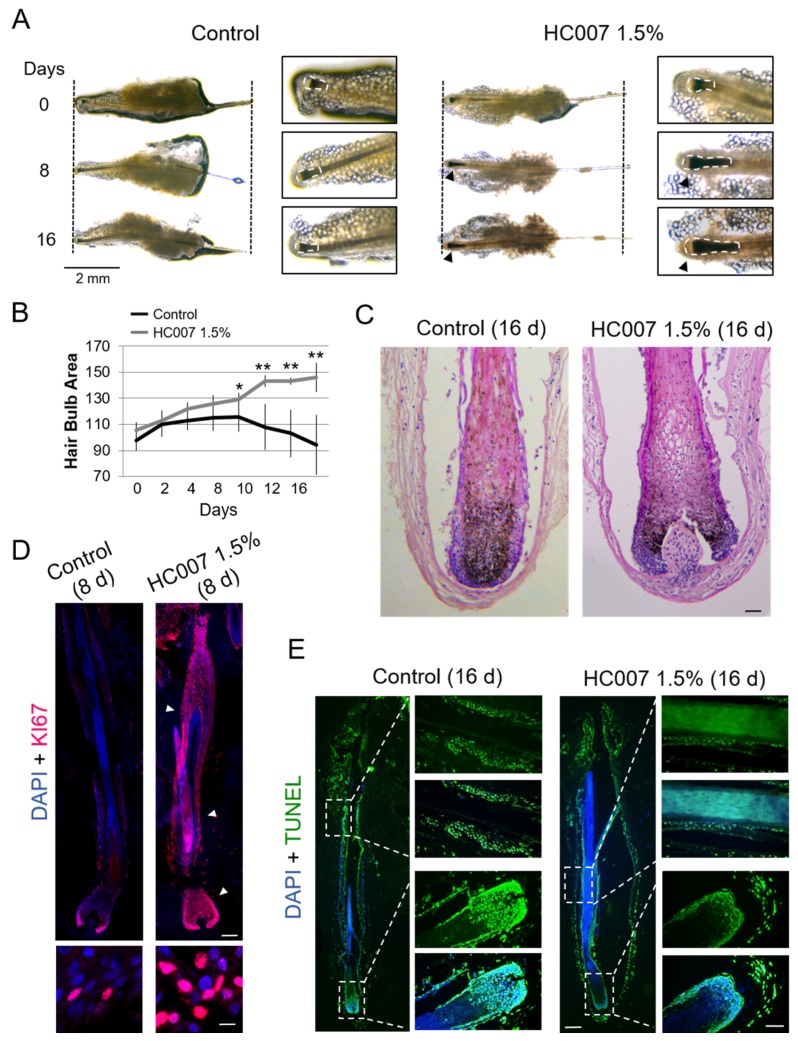
Sustained maintenance of human hair follicle growth ex vivo in a defined glycosaminoglycan hydrogel matrix (HC007). (**A**) Representative phase-contrast microscopy images of whole hair follicles showing significant thickening (black arrowheads) of the hair bulb area, defined here as an area encompassing hair bulb/dermal papilla and suprabulbar regions, after continuous growing in HC007 1.5%. Vertical dotted lines delimitate hair follicle length at time 0. A noticeable length increase in hair follicle length was observed in 50% of the HC007 1.5% samples. Enlarged images depicting typical hair bulb areas used in image quantification analysis (white dotted lines). Bar: 2 mm. (**B**) Time course quantification of hair bulb area in control and HC007 1.5% samples. Results are representative of at least 18 hair follicle units per condition. The mean +/− SD of *n* ≥ 18 for each experimental condition is represented and *t*-test was used for statistical analysis. *, significant *p* ≤ 0.1. **, significant *p* ≤ 0.05. (**C**) Representative histological sections stained with H&E of the hair bulb area in control and HC007 1.5% after 16 days in culture. Bar: 50 µm. (**D**) Confocal microscopy images (maximum projections) showing the localization of the cell proliferation marker KI67 in morphologically equivalent histological sections of hair follicles grown ex vivo for 8 days in control basal culture conditions or in HC007 1.5%. Bar: 100 µm. Enlarged images show nuclear KI67 staining in outer root sheath cells. Bar: 10 µm. (**E**) Identification of apoptotic cells by the TUNEL assay in morphologically equivalent histological sections of hair follicles grown ex vivo for 16 days in control basal culture conditions or in HC007 1.5%. Bars: 100 µm. Results shown in (**C**–**E**) are representative of at least 10 hair follicles in three independent experiments.

**Figure 2 ijms-20-01741-f002:**
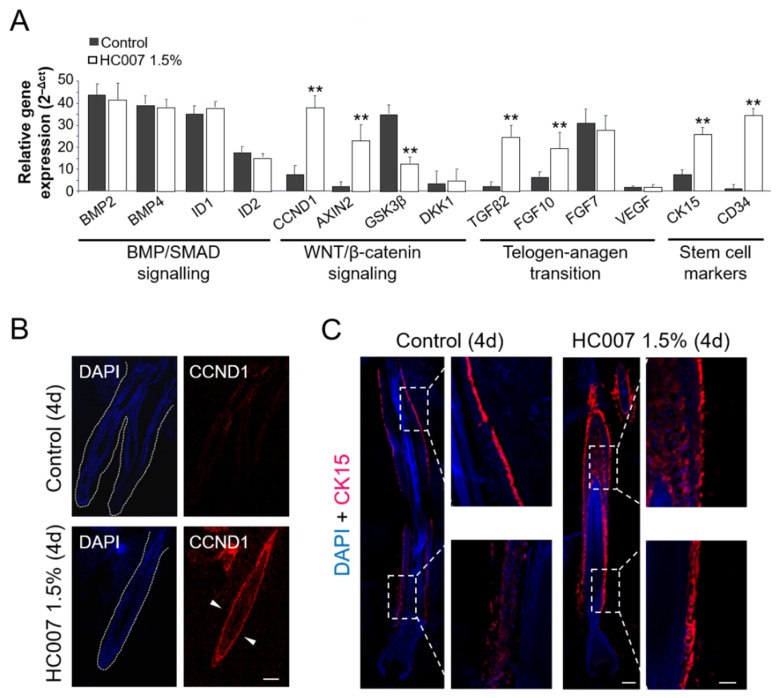
Activation of WNT/β-catenin signaling and of stem cell niche proliferation in human hair follicle growth ex vivo in a defined glycosaminoglycan hydrogel matrix (HC007). (**A**) Quantitative expression analysis by qRT-PCR analysis of selected genes. Input mRNA was obtained from three or four hair follicles per experimental condition, namely, control and HC007 1.5% samples, and the mean +/- SD of relative gene expression values, normalized to 18S rRNA, is represented. **, significant *p* ≤ 0.1. (**B**,**C**) Confocal microscopy images (maximum projections) of the immunolocalization of the (**B**) transcriptional target of WNT signaling CCDN1 (in red; white dotted lines delineate individual hair follicles; withe arrowheads indicate positive cells in the outer root sheath) and of the (**C**) CK15 stem cell marker (white dotted boxes are enlarged in right panels) in histological sections of human hair follicles grown ex vivo four days after HC007 1.5% treatments. DAPI was used as chromatin counterstain. Results shown in (**B**,**C**) are representative of at least 10 hair follicles in three independent experiments. Bars: 100 µm.

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
