# Peer review of "Sustained Human Hair Follicle Growth Ex Vivo in a Glycosaminoglycan Hydrogel Matrix"

_ijms, 2019, doi:10.3390/ijms20071741_

Round 1
Reviewer 1 Report
In this study, Fernández-Martos et al. use a GAG hydrogel (HC007) to mimic dermal matrix composition. Authors found out that HC007 ex vivo supports and promotes human hair follicle entrance into growing phase. The analysis of molecular mechanism indicate the contribution of WNT/β-catenin signaling determined by increased CCDN1 expression and reduced GSK3B expression.
This study is nicely addressed and point out about the potential use of GAG hydrogels to support ex vivo human hair follicle growth. In general, this work is acceptable for publication with some minor concerns:
-The hypothesis is very vague, the authors should define more accurately the work hypothesis.
-Maybe, authors can discuss about potential interplay between TGF-b2 and b-catenin signaling.
-Does possible that GSK3B inhibition enhances or mimic HC007 effects?
Author Response
REVIEWER 1
In this study, Fernández-Martos et al. use a GAG hydrogel (HC007) to mimic dermal matrix composition. Authors found out that HC007 ex vivo supports and promotes human hair follicle entrance into growing phase. The analysis of molecular mechanism indicate the contribution of WNT/β-catenin signaling determined by increased CCDN1 expression and reduced GSK3B expression.
This study is nicely addressed and point out about the potential use of GAG hydrogels to support ex vivo human hair follicle growth. In general, this work is acceptable for publication with some minor concerns:
-The hypothesis is very vague, the authors should define more accurately the work hypothesis.
We have now introduced in the Introduction section of the revised version a paragraph describing in more detail our working hypothesis:
“In this context, our aim in this work has been to investigate potential stimulatory effects of a defined GAG matrix on the hair follicle growth potential using ex vivo cultured human hair follicles as experimental system. In this biological model most cell types in the tissue rapidly cease to proliferate and enter into an apoptotic cell death program after serial culturing in basal conditions (e. g. no growth factors added to culture medium), therefore constituting an adequate system to evaluate the functional effects of GAGs on isolated mini-organs”.
-Maybe, authors can discuss about potential interplay between TGF-b2 and b-catenin signaling.
We have introduced in the revised version a paragraph discussing the potential interplay between TGF-b2 and b-catenin signalling:
“Moreover, it has been reported that WNT/β-catenin signaling during the onset of the anagen phase is accompanied by the specific expression of TGFβ2, that regulates proliferation, differentiation and extracellular matrix production of dermal fibroblasts [18]. The interplay between WNT/β-catenin and TGFβ/BMP/SMAD to regulate stem cell niche function has been also described in different adult tissues, including the hair follicle [19]. It can be hypothesized that the GAG matrix produced by dermal fibroblasts in response to TGFβ/BMP/SMAD signaling is an essential component in the regulation of the hair follicle cycle and, particularly, in the feed-back modulation of WNT/β-catenin signalling”.
-Does possible that GSK3B inhibition enhances or mimic HC007 effects?
This is a very interesting suggestion. Indeed, we are currently performing experiments using GSK3B and WNT inhibitors in this experimental system. We feel that this experiment is somewhat far of the scope of the present paper. It is our intention to report these results in a follow up more detailed paper.
Reviewer 2 Report
I am sorry but I feel that the authors have completely mis-understood the nature of a telogen follicle (by morphology) or the telogen to anagen transition. Thus the premise on which the manuscript is based needs to be re-examined.
Follicles termed 'telogen' are in fact in anagen. Effects see in the GAG matrix are effects associated with anagen maintenance in vitro, particularly failure of anagen maintenance in vitro, when the ORS cells to become more proliferative.
Author Response
REVIEWER 2
I am sorry but I feel that the authors have completely mis-understood the nature of a telogen follicle (by morphology) or the telogen to anagen transition. Thus the premise on which the manuscript is based needs to be re-examined.
Follicles termed 'telogen' are in fact in anagen. Effects see in the GAG matrix are effects associated with anagen maintenance in vitro, particularly failure of anagen maintenance in vitro, when the ORS cells to become more proliferative.
We totally agree with the comments of the reviewer. Therefore, we have completely revised the central premise of the manuscript. We now explicitly indicate that early anagen hair follicles have been actually used in this work and that while most hair follicles in basal conditions typically showed low cell proliferation rates followed by extensive induction of apoptosis in hair bulb and suprabulbar regions after serial ex vivo culture, growing hair follicles in HC007 1.5% promotes a sustained cell survival and maintenance of a highly proliferative phenotype in the hair follicle.
Reviewer 3 Report
The authors examined the effect of new-generated GAG hydrogel on human hair follicle growth. The manuscript raises too many questions:
1. It is not clear what control has been used in this experiment. There is no description of control samples in the section Results and Discussion. I was beginning to think that the comparison was made with the minus control (the same medium but without the active ingredient). But in the Methods section commercial GAG hydrogel is introduced as a control. The authors should mention about it in the Introduction, Result and Discussion sections. Moreover, with such a control all the conclusions seem inappropriate; extra details should be incorporated and all inconsistencies should be eliminated. If in reality the control was other than commercial GAG hydrogel, the author should clearly indicate it from the very beginning.
2. It is not clear what concentration of the GAG hydrogel was used. The data are inconsistent. The pictures indicate 1.5%, while the text of the article and Methods - 0.5%.
3. The manuscript claims analysis of hair follicles in the stage of telogen, but the pictures provided show anagen hair follicles.
4. Picture 1C is questionable, at least not informative. It seems that two different parts of the hair follicle are compared, because of the different size of the middle part of hair follicle. The whole hair follicle should be presented to address this issue.
5. Picture 2C is questionable as well. Specific for hair follicles CK15 staining is presented on follicles treated with HC007, while control follicles don’t have staining at all. In my extensive experience it seems very unlikely. To prove the reverse and support the conclusion about opposite pattern of CK15 staining, further experimental work is needed.
6. Materials and Methods section contains description of trichrome staining while the Result section doesn’t provide data on it.
7. Proofreading of text should be performed. (For example, sometimes signaling goes with double l).
Author Response
REVIEWER 3
The authors examined the effect of new-generated GAG hydrogel on human hair follicle growth. The manuscript raises too many questions:
1. It is not clear what control has been used in this experiment. There is no description of control samples in the section Results and Discussion. I was beginning to think that the comparison was made with the minus control (the same medium but without the active ingredient). But in the Methods section commercial GAG hydrogel is introduced as a control. The authors should mention about it in the Introduction, Result and Discussion sections. Moreover, with such a control all the conclusions seem inappropriate; extra details should be incorporated and all inconsistencies should be eliminated. If in reality the control was other than commercial GAG hydrogel, the author should clearly indicate it from the very beginning.
We thank the reviewer this accurate comment. Actually, Control samples have been grown in medium without HC007. The misunderstanding comes from a residual phrase from M&Ms section of previous papers. We have now checked and eliminate all inconsistencies.
2. It is not clear what concentration of the GAG hydrogel was used. The data are inconsistent. The pictures indicate 1.5%, while the text of the article and Methods - 0.5%.
Again, we thank the reviewer for noticing of this slip. The HC007 concentration in basal medium was 1.5%.
3. The manuscript claims analysis of hair follicles in the stage of telogen, but the pictures provided show anagen hair follicles.
As previously indicated, we totally agree with this comment. Therefore, we have completely revised the central premise of the manuscript. We now explicitly indicate that early anagen hair follicles have been actually used in this work and that while most hair follicles in basal conditions typically showed low cell proliferation rates followed by extensive induction of apoptosis in hair bulb and suprabulbar regions after serial ex vivo culture, growing hair follicles in HC007 1.5% promotes a sustained cell survival and maintenance of a highly proliferative phenotype in the hair follicle.
4. Picture 1C is questionable, at least not informative. It seems that two different parts of the hair follicle are compared, because of the different size of the middle part of hair follicle. The whole hair follicle should be presented to address this issue.
We have introduced new figures showing the distribution of KI67 whole hair follicles. As the results is not relevant with respect to main conclusions, we have eliminated comments and figures on the number of inner/outer root sheath cell layers to avoid further confusion.
5. Picture 2C is questionable as well. Specific for hair follicles CK15 staining is presented on follicles treated with HC007, while control follicles don’t have staining at all. In my extensive experience it seems very unlikely. To prove the reverse and support the conclusion about opposite pattern of CK15 staining, further experimental work is needed.
New figures showing the immunolocalization pattern of CK15 in whole hair follicles have now been introduced in the manuscript. In addition, further qRT-PCR analysis of CK15 and CD34 gene expression have been also introduced in the text.
6. Materials and Methods section contains description of trichrome staining while the Result section doesn’t provide data on it.
This section has been deleted at all.
7. Proofreading of text should be performed. (For example, sometimes signaling goes with double l).
We have performed extensive proof reading of the text.
Reviewer 4 Report
Studies in the current manuscript focused on the Induction of human hair follicle growth ex vivo in a glycosaminoglycan hydrogel matrix.
Paper is interesting but some corrections need to be addressed to strengthen the study:
1. Introduction
Authors should add and/or replace oldest references by new one. (PMID: 29495527, should be added)
2. Results and discussion
Concerning HC007, authors should precise why they used it at 1.5% (previous experiments?).
It should be of interest to study also FGF9/10 and to realize an immunolabeling of CD200.
In a general manner, authors should discuss in more detail the obtained results.
3. Materials and Methods:
Concerning the follicles: Authors have to precise the criteria of selection.
Concerning HC007, authors should provide mw od HA that they used.
Author Response
REVIEWER 4
Studies in the current manuscript focused on the Induction of human hair follicle growth ex vivo in a glycosaminoglycan hydrogel matrix.
Paper is interesting but some corrections need to be addressed to strengthen the study:
1. Introduction
Authors should add and/or replace oldest references by new one. (PMID: 29495527, should be added)
References have been update following the suggestions of the reviewer.
2. Results and discussion
Concerning HC007, authors should precise why they used it at 1.5% (previous experiments?).
Actually, HC007 1.5% concentration was choose based upon previous experimental work. This comment has been added to the test.
It should be of interest to study also FGF9/10 and to realize an immunolabeling of CD200.
We have now introduced the analysis of FGF10 gene expression, which is in fact induced upon growing in HC700 1.5%. We failed to detect FGF9 in our samples using three different PCR primer sets. Regarding CD200, we feel that the analysis of this protein, although potentially interesting, is far of the scope of this work.
In a general manner, authors should discuss in more detail the obtained results.
A more detailed description of results has been added to the text.
3. Materials and Methods:
Concerning the follicles: Authors have to precise the criteria of selection.
Criteria of selection has been added.
Concerning HC007, authors should provide mw od HA that they used.
References to previous work showing HA mw of HC007 has been included
Round 2
Reviewer 2 Report
The authors have re-evaluated the data and revised the original premise that the HC007 induced anagen from telogen in FUs. The data presented suggest HC007 can maintain anagen in FUs better than the basal supplemented WE medium.
The paper is improved. However, I suggest it is critically read by a knowledgable hair biology colleague before resubmission.
Specific comments
Line 54... Important roles for these molecules have been postulated during 55 skin regeneration and wound healing and in the process of skin aging [2,3,5]. However, little is 56 actually known at a functional level about the physiological implication of GAGs in the regulation of 57 the hair follicle growth cycle
The evidence for role of GAGs in hair growth is actually well known...Authors should read and cite the work of Couchman J. and or Westgate GE and or Taylor A....(https://www.ncbi.nlm.nih.gov/pubmed/1610689, https://www.ncbi.nlm.nih.gov/pubmed/1704038, https://www.ncbi.nlm.nih.gov/pubmed/7665924).
Bulb area: without any reference to what this means, it is hard to interpret Line 79/88 where the authors describe a thickening of the hair bulb in the legend of Fig, 1 and show this graphically. Why does the bulb area grow?. Do the bulbs actually thicken? can the authors describe what they mean by thickening - ie is this increase in diameter, more cells, more ECM? The authors could show images of the bulb region of control and treated follicles showing how bulb area is calculated? This is also important for judging anagen dermal papilla and bulb morphology, as this is a key end point measure for the action of HC007..
Line 106 - referring to BMP expression. The graph in Fig 2A shows no significant difference between control and treated, therefore the authors cannot make this statement 'mimicking a telogen-like phenotype'. or 'Interestingly, this 109 expression pattern was not affected by HC007 treatments (Fig. 2A)'.????
Line 140: all along the inner root sheath in HC007....activation of the bulge stem cell niche
In anagen the IRS is non proliferative...??? please check the anatomy carefully. ORS cell proliferation is known to increase in vitro.... The authors cannot conclude activation of the stem cell niche - possibly maintenance??
Line 142: The results reported here indicate for the first time that a defined GAG matrix can promote and 143 support the sustained growth of human telogen hair follicles ex vivo.
Sustained growth of human anagen hair follicles.......
Can the authors also make reference to Matrigel (a similar GAG containing substrate) which has been used to support dermal papilla cells for hair inductive properties. see recent paper https://www.ncbi.nlm.nih.gov/pmc/articles/PMC4161057/
Would the authors like to comment on whether HC007 has benefits in hair transplantation or other hair regenerative processes whether used ex vivo or in vivo as part of the transplantation?
Methods
line 71-2 and 175/176....Selected FUs encompassed for the most part hair follicles in early anagen or the late telogen/anagen transition.
I would argue that it is not possible to determine early anagen or late telogen to anagen transition in FU. The Follicles observed in the figures all look like anagen VI. The authors should demonstrate the stage they describe with histology or concede that follicles are in Anagen VI - albeit it is impossible to know how long any follicle has been in anagen VI...
line 196 says culture was for 4 days, Figure 1 suggest up to 16 days? Authors should correct the description of the methods.
Line 177. Williams E medium (Sigma) supplemented with 10x Penicilin/Streptomycin, 10x Amphotericin B, 178 and 2 mM L-Glutamin (all from Gibco), 5 µg/ml Insulin, 5 µg/ml transferring, 20 pM T3 hormone, 0.083 µg/ml cholera toxin and 0.4 µg/ml hydrocortisone at 37°C in a 5% CO2 humidified atmosphere
Can the authors explain the use of this medium composition? why include T3, Transferrin and cholera toxin? The work of Westgate et al defined the optimal medium for isolated hair growth. Do FUs need these other additions? (https://www.ncbi.nlm.nih.gov/pubmed/7692925)
Line 192. fresh medium containing......?
Author Response
Specific comments
Line 54... Important roles for these molecules have been postulated during 55 skin regeneration and wound healing and in the process of skin aging [2,3,5]. However, little is 56 actually known at a functional level about the physiological implication of GAGs in the regulation of 57 the hair follicle growth cycle
The evidence for role of GAGs in hair growth is actually well known...Authors should read and cite the work of Couchman J. and or Westgate GE and or Taylor A....(https://www.ncbi.nlm.nih.gov/pubmed/1610689, https://www.ncbi.nlm.nih.gov/pubmed/1704038, https://www.ncbi.nlm.nih.gov/pubmed/7665924).
Actually, the references mentioned by the reviewer does not support or demonstrate (perhaps suggest) any functional role of GAGs in hair growth. Two of them (Westgate et al. J. Invest. Dermatol. 96:191, 1992; du Cros et al. J. Invest. Dermatol. 105:426, 1995) are just inmmunohistochemical studies showing the localization and expression levels of different proteoglycans (not GAGs) during the hair follicle cycle in mammalian skin. The other one is just an analysis of GAG synthesis in cultured human hair follicle dermal papilla cells as compared with non-follicular dermal fibroblasts. In our opinion these cannot be considered as evidences for the role of GAGs in hair growth. Due to space restriction we will no reference these works in our manuscript. To our knowledge, functional evidences suggesting a direct functional implication of GAGs in hair growth are very scarce.
Bulb area: without any reference to what this means, it is hard to interpret Line 79/88 where the authors describe a thickening of the hair bulb in the legend of Fig, 1 and show this graphically. Why does the bulb area grow? Do the bulbs actually thicken? can the authors describe what they mean by thickening - ie is this increase in diameter, more cells, more ECM? The authors could show images of the bulb region of control and treated follicles showing how bulb area is calculated? This is also important for judging anagen dermal papilla and bulb morphology, as this is a key end point measure for the action of HC007.
Here we have defined the hair bulb area as an area encompassing de hair bulb/dermal papilla and the suprabulbar region as depicted in new Fig 1. This area increases in cell numbers and, hence in diameter after ex vivo culture in HC007.
Line 106 - referring to BMP expression. The graph in Fig 2A shows no significant difference between control and treated, therefore the authors cannot make this statement 'mimicking a telogen-like phenotype'. or 'Interestingly, this 109 expression pattern was not affected by HC007 treatments (Fig. 2A)'.????
We have deleted these confusing statements.
Line 140: all along the inner root sheath in HC007....activation of the bulge stem cell niche
In anagen the IRS is non proliferative...??? please check the anatomy carefully. ORS cell proliferation is known to increase in vitro.... The authors cannot conclude activation of the stem cell niche - possibly maintenance??
Thanks for the annotation. Indeed, we were actually referring to ORS and we have consequently corrected this issue. On the other hand, our new qRT-PCR data show a very significant increase of CK15 and CD34 stem cell marker expression clearly suggesting an activation of the stem cell niche.
Line 142: The results reported here indicate for the first time that a defined GAG matrix can promote and 143 support the sustained growth of human telogen hair follicles ex vivo.
Sustained growth of human anagen hair follicles.......
Thanks for notice this slip. We have consequently corrected the word.
Can the authors also make reference to Matrigel (a similar GAG containing substrate) which has been used to support dermal papilla cells for hair inductive properties. see recent paper https://www.ncbi.nlm.nih.gov/pmc/articles/PMC4161057/
Matrigel is a reconstituted basement membrane preparation that is extracted from the Engelbreth-Holm-Swarm (EHS) mouse sarcoma, a tumor rich in extracellular matrix proteins. This material, once isolated, is approximately 60% laminin, 30% collagen IV, and 8% entactin. Although it is true that Matrigel also contains a small proportion of the heparan sulfate proteoglycan perlecan, in our opinion, and taking into account our space restrictions, this fact does not justify the inclusion of this reference in the manuscript.
Methods
line 71-2 and 175/176....Selected FUs encompassed for the most part hair follicles in early anagen or the late telogen/anagen transition.
I would argue that it is not possible to determine early anagen or late telogen to anagen transition in FU. The Follicles observed in the figures all look like anagen VI. The authors should demonstrate the stage they describe with histology or concede that follicles are in Anagen VI - albeit it is impossible to know how long any follicle has been in anagen VI...
To avoid further misunderstandings, we now state in the manuscript that “selected FUs encompassed for the most part hair follicles in the growing (anagen) phase”.
line 196 says culture was for 4 days, Figure 1 suggest up to 16 days? Authors should correct the description of the methods.
We have corrected this inconsistency.
Line 177. Williams E medium (Sigma) supplemented with 10x Penicilin/Streptomycin, 10x Amphotericin B, 178 and 2 mM L-Glutamin (all from Gibco), 5 µg/ml Insulin, 5 µg/ml transferring, 20 pM T3 hormone, 0.083 µg/ml cholera toxin and 0.4 µg/ml hydrocortisone at 37°C in a 5% CO2 humidified atmosphere
Can the authors explain the use of this medium composition? why include T3, Transferrin and cholera toxin? The work of Westgate et al defined the optimal medium for isolated hair growth. Do FUs need these other additions? (https://www.ncbi.nlm.nih.gov/pubmed/7692925)
The use of cholera toxin and transferrin for sustained mammalian keratinocyte culture in serum free conditions has been reported (The Keratinocyte Handbook. Cambridge University Press 1995; Okada et al. J. Invest. Dermatol. 79:42, 1982; Batista et al. J. Stem Cell Res. Ther. 1:101, 2010). The use of T3 to expand anagen in ex vivo human hair follicle culture has been also reported (van Breek et al. J. Clin. Endocriol. Metab. 93:4381, 2008).
Line 192. fresh medium containing......?
Thanks for the annotation. We have corrected this paragraph.
Reviewer 3 Report
The new version of paper is improved, but still raises some questions.
The ex vivo organ hair follicle culture is a classical model to assess hair growth. It is interesting whether the authors checked the influence of HCOO7 on the hair growth (elongation of hair shaft). It would be the best demonstration of sustaining of hair growth by HCOO7. Particularly taking into account the raise of ki67-positive cells, stated by the authors.
The authors declare that the thickening of the hair bulb and suprabulbar regions efficiently support sustained hair follicle growth (83-84 lines). The conclusion is not evident for me. How the thickening of the bulb can contribute to hair growth? What contributed to the thickening, cell proliferation or matrix synthesis? Or can the thickening just result from the tissue swelling? Should be discussed more and preferably proved by references.
Picture 1B. The quality of picture does not allow distinguishing if Ki67 is located in nuclei. Could you please show the enlarged picture B (HCOO7).
Author Response
The new version of paper is improved, but still raises some questions.
The ex vivo organ hair follicle culture is a classical model to assess hair growth. It is interesting whether the authors checked the influence of HCOO7 on the hair growth (elongation of hair shaft). It would be the best demonstration of sustaining of hair growth by HCOO7. Particularly taking into account the raise of ki67-positive cells, stated by the authors.
Actually, we have observed a clearly noticeable, although hardly measurable, increase in the length of the hair shaft in a considerable proportion (about 50%) of hair follicles grown in HC007 1.5%. We have added this observation to text and new Fig. 1A.
The authors declare that the thickening of the hair bulb and suprabulbar regions efficiently support sustained hair follicle growth (83-84 lines). The conclusion is not evident for me. How the thickening of the bulb can contribute to hair growth? What contributed to the thickening, cell proliferation or matrix synthesis? Or can the thickening just result from the tissue swelling? Should be discussed more and preferably proved by references.
The enlargement of the hair bulb area associated with hair follicle growth and overall increase in cell numbers is well reported in the literature (see for example Oh et al. J. Invest Dermatol 136:34, 2016 and references therein). In this revised version, we provide new histological images (new Fig 1C) showing a clear overall increase in cell numbers in hair follicles grown in HC007 1.5% for 16 days as compare to controls. We are aware that these are not the best histological sections ever, but the only available in the due time granted for reply and we honestly believe that are representative enough. We can provide better images if more time for reply is granted. In any case, we accept and thank the very pertinent comment of the reviewer indicating that, in the experimental approach of this short communication, it cannot be directly inferred than an enlargement of hair bulb area is directly associated with hair follicle growth. In consequence, we have modified the annotated statement, focusing on a cell proliferation phenotype instead of hair follicle growth.
Picture 1B. The quality of picture does not allow distinguishing if Ki67 is located in nuclei. Could you please show the enlarged picture B (HCOO7).
We have now incorporated new enlarged images showing the nuclear localization of KI67 in outer root sheath cells (New Fig 1D).
Round 3
Reviewer 2 Report
Thank you to the authors for addressing the reviewers comments. In relation to adding suggested references, if space is limited, then additional references may not be possible; however, referring to older literature is valuable to the readers who may not be aware of the work done in the past when the actual relationship (even if not causal) between GAGs and hair growth was first described together with reference cited..
The amendment to Figure 1 using dotted lines to delineate an area measured seems rather arbitrary and I still am not convinced that the methods used to generate the data in Fig 1 B are sufficiently described to support the statement page 2 line 83/84 below...
By contrast, growing hair follicles in HC007 1.5% promoted a sustained and statistically significant thickening of the hair bulb and suprabulbar regions (Fig. 1A-C)
Surely the bulb is not actually thickening beyond its size at isolation? For proving this, the authors need to demonstrate a more detailed and accurate method of bulb area definition with clear anatomical criteria and images.
As with the amended title of the paper, the evidence supports that HC007 helps maintenance of hair follicle anagen morphology ex vivo. The authors could also suggest that researchers working on the ex vivo HF model may like to consider HC007 as a very useful addition to the media such that HF functions are maintained for longer. This would be helpful to this community.
Author Response
In relation to adding suggested references, if space is limited, then additional references may not be possible; however, referring to older literature is valuable to the readers who may not be aware of the work done in the past when the actual relationship (even if not causal) between GAGs and hair growth was first described together with reference cited.
In the Introduction section we now indicate that “despite different reports showing a cyclic expression of GAGs and different proteoglycans during the hair follicle growth cycle [3,5,6], little is actually known at a functional level about the physiological implication of GAGs in the regulation of this biological oscillator”.
The amendment to Figure 1 using dotted lines to delineate an area measured seems rather arbitrary and I still am not convinced that the methods used to generate the data in Fig 1 B are sufficiently described to support the statement page 2 line 83/84 below...
By contrast, growing hair follicles in HC007 1.5% promoted a sustained and statistically significant thickening of the hair bulb and suprabulbar regions (Fig. 1A-C)
Surely the bulb is not actually thickening beyond its size at isolation? For proving this, the authors need to demonstrate a more detailed and accurate method of bulb area definition with clear anatomical criteria and images.
We have now defined, in Results/Discussion and Material/Methods sections, the hair bulb area as the hair follicle territory encompassing the hair bulb and dermal papilla as well as the most intensively pigmented area of the suprabulbar region, including associated inner and outer root sheets. This area is characterized by high cell proliferation and melanogenesis during the anagen phase of the hair follicle cycle. We have added an additional reference to support this perspective. (Slominsky et al. J Invest Dermatol, 2005 and references therein). In addition, and to avoid further confusion we have substituted “thickening” by “enlargement”.
As with the amended title of the paper, the evidence supports that HC007 helps maintenance of hair follicle anagen morphology ex vivo. The authors could also suggest that researchers working on the ex vivo HF model may like to consider HC007 as a very useful addition to the media such that HF functions are maintained for longer. This would be helpful to this community.
We thank the reviewer this interesting observation. We have added such accurate suggestion and the end of the Discussion.